# Theoretical Investigation of Skyrmion Dynamics in Pt/Co/MgO Nanodots

**DOI:** 10.3390/ma15217474

**Published:** 2022-10-25

**Authors:** Djoudi Ourdani, Mohamed Belmeguenai, Mihai Gabor, Andrey Stashkevich, Yves Roussigné

**Affiliations:** 1LSPM (CNRS-UPR 3407), 99 Avenue Jean-Baptiste Clément, Université Paris 13, 93430 Villetaneuse, France; 2Center for Superconductivity, Spintronics and Surface Science, Physics and Chemistry Department, Technical University of Cluj-Napoca, Memorandumului No. 28, RO-400114 Cluj-Napoca, Romania

**Keywords:** spintronics, skyrmion stability, nanodots, magnetization dynamics

## Abstract

In this article, we present a numerical study on stabilization and eigenmodes of the so-called skyrmion chiral spin texture in nanometric dots. The first aim of this study is to identify the appropriate multilayer in a set of Pt/Co/MgO structures with different Co thicknesses that have been previously experimentally characterized. Stabilization occurs if the energy favoring skyrmions is greater than the geometric mean of the exchange and anisotropy energies. Both the energy favoring skyrmions and the anisotropy contribution depend on the Co thickness. The appropriate multilayer is obtained for a specific Co thickness. MuMax simulations are used to calculate the precise static magnetization configuration for the experimental parameters, allowing us select the appropriate structure. Moreover, in view of experimental study of skyrmion dynamics by means of Brillouin light scattering, the eigenfrequency, eigenmode profile, and spectral density are calculated for different dot sizes. Finally, the optimal dot size that allows for a feasible experiment is obtained.

## 1. Introduction

Information coding via magnetic state is becoming one of the leading information storage technologies. This idea dates back to the 1960s and 1970s, when cylindrical bubble domains in micrometric ferrite films were intensively investigated with the aim of creating non-volatile magnetic computer memory using each bubble domain of micrometric size as one bit of data [1]. Half a century later this fruitful idea has been revisited, backed by a major breakthrough in ultra-thin film technologies, as well as considerable progress in the solid-state physics of ferromagnetic (FM) chiral nanostructures such as skyrmions or merons. Theoretically predicted in 2006 [2] as topologically protected local whirls of the spin configuration, these have proved to be particularly stable. Contrary to the micrometric bubble domains of the 1970s, which were stabilized predominantly by dipole–dipole interactions (DDIs), the nanometric chiral analogues of today [3,4,5] can only be stabilized in the presence of anti-symmetric Dzyaloshinskii–Moriya interactions (DMI) [6,7]. Application-wise, these are the skyrmions hosted in ultra-thin FM films (1–5 nm) in contact with a heavy metal (HM), and are regarded as promising candidates for the role of information carriers in next-generation magnetic computing and logic systems such as racetrack memories [8]. Moreover, these small quasi-particles (down to the inter-atomic scale [9,10,11,12,13]) have demonstrated drastically enhanced velocity, and very small current densities are needed to displace them. This is theoretically explained by stabilization of ρ-skyrmions, within which the core and peripheries are separated by a Néel domain wall (DW), which is favored by iDMI and experimentally exploited in applications relying on DW motion. No less important, it has been shown that the iDMI properties of the iDMI FM/HM interface can be controlled electrically [14], an important step towards energy-efficient spintronic computing technologies. At the same time, a novel trend involving the employment of another type of skyrmionic dynamics based on high-frequency oscillations is gaining momentum. This rapidly developing research area holds undeniable promise for important applications in the emerging field of microwave spintronics such as high-frequency signal treatment, including neuromorphic computing. Skyrmion-related functionalities basically fall into two main categories, namely, skyrmion-based magnonic crystals (MC) [15,16,17] and effective nano-sized emitters of ultra-short exchange spin waves (SWs) [18,19]. Skyrmion MCs feature unique versatility; their dispersion characteristics, primarily bandgaps, can be tailored for a specific application by a proper choice of the skyrmion array geometry [20,21], and can be further tuned “dynamically” by external fields, electric currents, or spin waves in so-called “dynamic magnonic crystals” [22,23]. On the other hand, skyrmions and spin helices in general are regarded as promising candidates for eliminating a critical bottleneck in nanomagnonics and its applications, potentially replacing costly and technologically challenging lithographic nano-antennae as emitters of short nanometric SWs. Moreover, a recent publication has mentioned the possibility of using ultra-small atomic scale skyrmions as q-bits for Quantum Information applications [24]. A combination of two skyrmion “breathing” eigenmodes is predicted to be a physical realization of a q-bit state.

In the theoretical research carried out over the last decade a number of different methods have proven to be effective, especially when cleverly combined. Thus, in the early paper by Rohart et al. [25], an adequate analytical description of the phenomena underlying the skyrmion stabilization process was reported. The authors adopted a seemingly reasonable approximation in which dipolar coupling is not considered directly; instead, *K* is regarded as an effective anisotropy constant expressed as a difference between the magnetocrystalline and shape contributions, the latter taking into account dipole interactions ($K=K_{MC}-2\pi M^{2}$, where $K_{MC}$ is the magneto crystalline anisotropy constant and *M* is the magnetization at saturation). In turn, the skyrmion stability depends on a figure of merit $\left|D\right|/\sqrt{A|K|}$ involving the iDMI strength (*D*), exchange constant (*A*), and effective anisotropy (*K*). According to this approach, a skyrmion can be stable only if the negative iDMI contribution to the Néel energy prevails over the positive contribution due to the Heisenberg exchange. In the present work, we study the stabilization of skyrmions in dots and their dynamic high-frequency eigenmodes by revisiting the skyrmion stability criteria. In layered structures with realistic magnetic parameters, the above-mentioned “effective shape anisotropy” approximation is no longer valid, and dipole interactions should be rigorously taken into account, first statically and then dynamically. To this end, we adopt the following strategy. Because *D* and *K* largely depend on the thickness of the magnetic film, the film thickness is carefully chosen prior to any measurements on skyrmions, allowing the stabilization criterion to be verified. The MuMax code [26] can correctly take into account dipole fields, and is chosen as the working tool for numerical simulations on the basis of the analytic calculations of SW modes by Z. Gareeva et al. [27]. Special attention is paid to the issues related to optimizing the experimental procedure. As the dot size strongly influences the skyrmion dynamics, the dependencies of the eigenfrequencies on this parameter need to be estimated to ensure that the investigated spectra lie within the range reachable by experimental techniques such as Brillouin light scattering. In addition, the skyrmion-related signal should be strong enough to be detectable. All these issues are discussed in the following sections.

## 2. Samples and Calculations

### 2.1. Samples

In this section, we recall the measured properties on a series of samples [28] where a specific structure has been identified as a good candidate for hosting skyrmions. The structure of the samples is Pt(1 nm)/Co(t)/MgO(1nm), where t is varied from 0.8 to 5.8 nm. The respective Co thickness dependence of |D| and *K* are shown in the inset of Figure 1. Each sample is represented by a point in the ($\sqrt{A|K|}$, |D|) diagram shown in Figure 1. We deduce from this experimental study that the appropriate Co thickness is t = 1.4 nm, for which $\left|D\right|/\sqrt{A|K|}$ is close to 1, as |D| is the largest and |K| is the smallest. The magnetic parameters for this sample are A=1.6 μerg/cm, M=1220 emu/cm${}^{3}$, D=−1.34 erg/cm${}^{2}$, and K=−1.04 Merg/cm${}^{3}$. Thus, these parameters are considered for numerical calculations.

### 2.2. Calculations

We used the free micromagnetic software MuMax [26] to perform static and dynamic calculations. These calculations rely on the energy per volume unit $E_{v}=-\vec{H}.\vec{M}-K_{MC}{\left(M_{z}/M\right)}^{2}-\frac{1}{2}{\vec{H}}_{dem}.\vec{M}+\left(D/M^{2}\right)\left(\vec{M}.\nabla M_{z}-\left(\nabla .\vec{M}\right)M_{z}\right)+\left(A/M^{2}\right)\left({\left(\nabla M_{x}\right)}^{2}+{\left(\nabla M_{y}\right)}^{2}+{\left(\nabla M_{z}\right)}^{2}\right)$, where the z axis is perpendicular to the magnetic layer, the x and y axes are in-plane, $\vec{M}$ is the magnetization, $K_{MC}$ is the anisotropy energy (regardless of shape anisotropy), *A* is the exchange constant, *D* is the effective DMI constant, $\vec{H}$ is the external field, and ${\vec{H}}_{dem}$ is the demagnetizing field. Here, ${\vec{H}}_{dem}$ is determined from the Maxwell equations $\nabla .({\vec{H}}_{dem}+4\pi \vec{M})=0$, $\nabla \times {\vec{H}}_{dem}=\vec{0}$ and the boundary conditions. For the specific material parameters corresponding to the experimental results, we studied the static configuration for different dot diameters in the range of 100–300 nm. The scheme of calculations is described in Figure 2; starting from the out-of-plane saturated configuration, the static configuration is obtained by energy minimization; this configuration is then disturbed and the response is analyzed to obtain the eigenfrequency and the corresponding profile. With the parameters (measured on real samples) and considering the experimental protocol where the sample is initially saturated out-of-plane, only one configuration can be stabilized. Other configurations might exist, as discussed in [29], and this could be an avenue of investigation in future research; however, the aim of the paper is to prepare an experimental study. In this study, the sample is initially saturated out-of-plane, ensuring that the magnetization configuration is known.

Two examples of static configurations corresponding to opposite effective anisotropies are shown in Figure 3. The calculated magnetization configurations for different dot sizes with the previously determined parameters are shown in Figure 4. The interaction between dots in a lattice is shown in Figure 5. The static configuration being identified, we then analyze the dynamic response to a small out-of-plane magnetic pulse. The Fourier analysis of the response provides the spectral density along with both the eigenfrequencies (positions of peaks) and corresponding amplitudes (heights of peaks). The calculated spectra are shown in Figure 6. Variations of the eigenfrequencies with the dot radius are shown in Figure 7. For each eigenfrequency, the mode profile is calculated the using Joo Von Kim method [30]. The calculated profiles are shown in Figure 8. Finally, we calculate the response to an in-plane magnetic field pulse in order to evidence modes with azimuthal profiles.

## 3. Results and Discussion

### 3.1. Static Configuration

First of all, the criterion for skyrmion stability is discussed. MuMax calculations use the energy with no approximation, and therefore the discussion is based on MuMax simulations. Nevertheless, to have physical insight, we have used the Rohart et al. criterion, where the demagnetizing contribution is approximated by a shape anisotropy, i.e., $-\left(1/2\right){\vec{H}}_{dem}.\vec{M}$ is replaced by $2\pi {\left(M_{z}\right)}^{2}$. The same approach is used in [29] and yields the same criterion (equation 14 in [29] and critical DMI energy consant $D_{c}$ in [25]). This rough calculation of the demagnetizing contribution allows a simple criterion to be derived, which we use as a first guess to select one sample hosting skyrmions. Then, the real magnetization configuration is calculated with the real demagnetizing field derived from Maxwell’s equations. Numerical calculation does not allow a skyrmion in a dot when the effective anisotropy is positive; in Figure 3, we present two cases with the same effective anisotropy in terms of the absolute value |K|=1.04 Merg/cm${}^{3}$ for a dot diameter of 150 nm. For a negative value (corresponding to spontaneous in-plane magnetization for a film), the relaxed configuration from out-of-plane saturated magnetization is a skyrmion, while for a positive value (corresponding to spontaneous perpendicular magnetization for a film) the final configuration remains mainly out-of-plane. This discrepancy with the criterion is probably due to the approximation used to estimate the demagnetizing effect. This effect has been better taken into account by Bernand-Mantel et al. [31]. Moreover, if K>0, a low value of *D* is not sufficient to allow a domain wall. On the other hand, for K<0, the magnetization tends to be in-plane inside the dot, while the demagnetizing field combined with DMI tends to make the magnetization out-of-plane in the vicinity of the dot edge. Here, we rely mainly on MuMax calculations. The simple Rohart et al. criterion is a first approximation. For a negative value of *K* (such as that chosen among the experimental parameters), the numerical calculations allow a skyrmion configuration. Nevertheless, such a configuration is allowed only for a dot diameter at least equal to 110 nm (see Figure 4). This is the effect of the energy cost for a circular domain wall. Consequently, the static configuration for a dot diameter of 100 nm shows only a straight domain wall for which the length is reduced. In order to check whether dipolar interaction between skyrmions substantially disturbs the static configuration, we consider a lattice of 110 nm broad dots separated by 110 nm wide pitches (see Figure 5). These parameters are technically feasible. The calculated configuration is hardly affected by interactions between dots. In the following, only one dot is considered, even though in the BLS experiments an assembly of dots is probed.

**Figure 3 materials-15-07474-f003:**
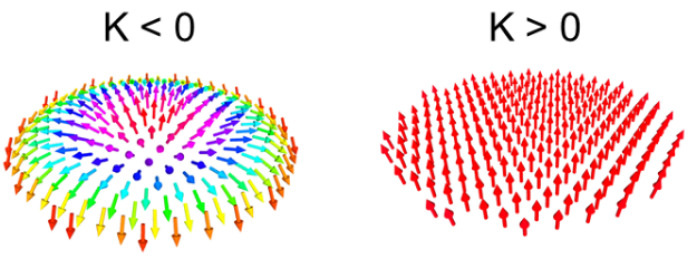
Static configurations for nanodot of 150 nm diameter in the case of K<0 (left) and K>0 (right). K<0 refers to spontaneous in-plane magnetization for a film. K>0 refers to spontaneous perpendicular magnetization for a film. The calculation parameters are: dot diameter = 150 nm, dot thickness = 1.4 nm, M=1220 emu/cm${}^{3}$, |K|=1.04 erg/cm${}^{3}$, D=−1.34 erg/cm${}^{2}$, A=1.6
μerg/cm.

**Figure 4 materials-15-07474-f004:**
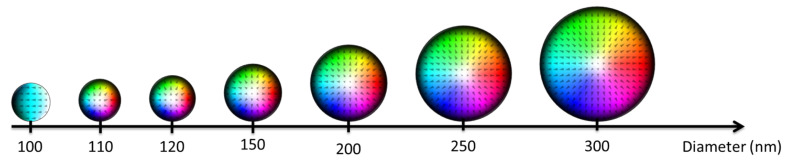
Static configurations for various diameters of nanodots. The calculation parameters are: dot thickness = 1.4 nm, M=1220 emu/cm${}^{3}$, K=−1.04 erg/cm${}^{3}$, D=−1.34 erg/cm${}^{2}$, A=1.6
μerg/cm.

**Figure 5 materials-15-07474-f005:**
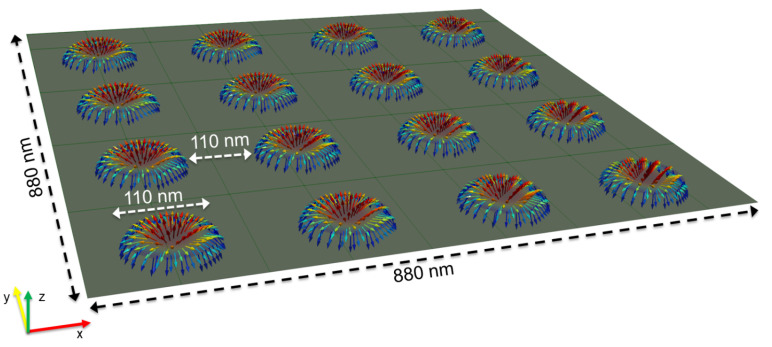
Lattice of skyrmions obtained after patterning. The calculation parameters are: dot thickness = 1.4 nm, M=1220 emu/cm${}^{3}$, K=−1.04 erg/cm${}^{3}$, D=−1.34 erg/cm${}^{2}$, A=1.6
μerg/cm.

### 3.2. Eigenmodes

The calculated spectrum shown in Figure 6 is obtained for a pulse applied field $H_{0}\sin\left(2\pi f_{0}t\right)/\left(2\pi f_{0}t\right)$ with $H_{0}=5$ Oe and $f_{0}=50$ GHz along the dot normal. The amplitude $H_{0}$ is chosen to slightly disturb the static configuration. The frequency $f_{0}$ defines the frequency range. The dot diameter is 150 nm. The spectrum reveals a series of peaks at 0.67, 4.50, 10.15, 18.47, 29.28, and 42.60 GHz. The nature of the corresponding modes is discussed later, after calculating the profiles. Nevertheless, the pulse being uniform and out-of-plane, we expect modes with only radial dependence. Moreover, Heisenberg exchange increases the frequency for faster variation along the radius. Consequently, the lowest frequencies correspond to the most uniform profiles. As a matter of fact, the frequency 0.67 GHz is associated with the skyrmion motion described by the Thiele equation [32]. As it is off the measurable frequency range by means of Brillouin light scattering, we use the second mode as a reference for intensity comparison. The spectral density enables estimation of the time required to obtain a measurable signal by Brillouin light scattering. For an array of saturated dots, the required time is on the order of 1 h. The time needed for the reference mode is expected to be similar. According to the calculations shown in Figure 6, the 10.15 GHz mode amplitude is the third of the reference modes. Thus, this mode requires 9 h of accumulation. The 18.47 GHz mode amplitude is the ninth of the reference modes, and the required accumulation time is 81 h.

As shown in Figure 7, the eigenfrequencies decrease with the diameter. This can be explained by Heisenberg exchange; increasing the diameter means slowing the variation along the radius, and thus reducing the exchange contribution. The curves plotted in Figure 7 correspond to the linear dependence with the square of the inverse diameter. The numerical trends are well fitted by this linear dependence, as expected from exchange contribution. Due to instrumental limitation of the Brillouin light scattering technique, the frequency has to be greater than 3 GHz. Moreover, the signal decreases with the rank of the mode. Thus, the dot with diameter 150 nm seems a good candidate for measurements, as the second mode frequency is 4.5 GHz; a larger diameter puts the second mode frequency off the measurable frequency range, while a lower diameter is too close to the critical diameter for allowing a skyrmion.

**Figure 6 materials-15-07474-f006:**
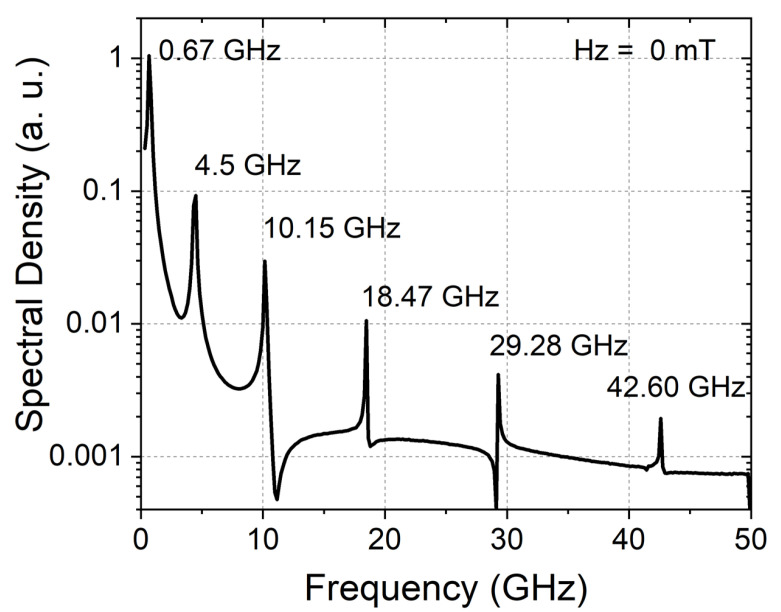
Spectral density in zero static field obtained from Fourier analysis of the dot response calculated by MuMax with the following parameters: dot diameter = 150 nm, dot thickness = 1.4 nm, M=1220 emu/cm${}^{3}$, K=−1.04 erg/cm${}^{3}$, D=−1.34 erg/cm${}^{2}$, A=1.6
μerg/cm.

**Figure 7 materials-15-07474-f007:**
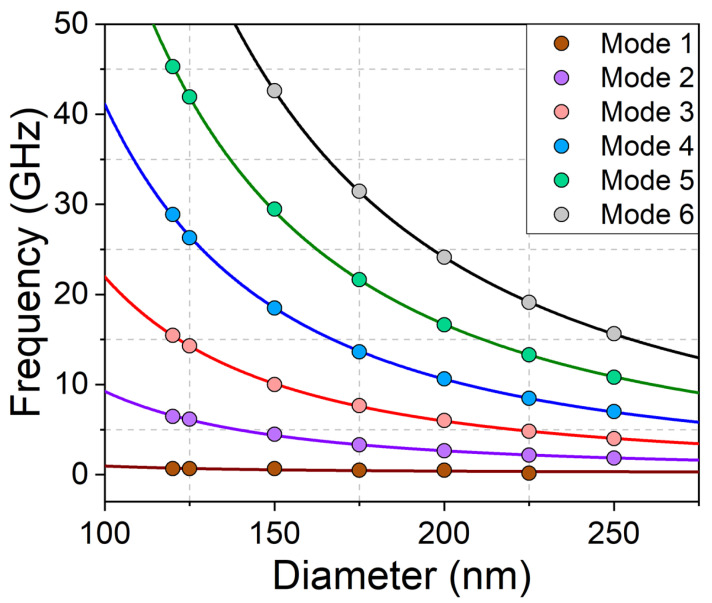
Eigenfrequencies variations with dot diameter. The calculation parameters are: dot thickness = 1.4 nm, M=1220 emu/cm${}^{3}$, K=−1.04 erg/cm${}^{3}$, D=−1.34 erg/cm${}^{2}$, A=1.6
μerg/cm.

Profiles corresponding to the eigenfrequencies determined from the spectrum in Figure 6 are shown in Figure 8. For the sake of clarity, three presentations are proposed: a profile of the out-of-plane magnetization component along a diameter (1D representation), mapping of this component using a color code (2D representation), and 3D representation. These profiles mainly depend on the distance from the centre. The slight azimuthal variations are numerical artifacts; as the disturbing field does not depend on the azimuthal angle, the response is not expected to present such a dependence. Real azimuthal modes excited by an appropriate disturbing field are discussed later. As expected, the more oscillations of the magnetization, the higher frequency because of exchange energy.

Calculations for modes excited by an in-plane field are shown in the two last lines of Figure 9. The in-plane exciting field breaks the initial symmetry. Consequently, the profiles depend on both the distance to the dot centre and on the azimuthal angle. Interestingly, these modes are observable by Brillouin light scattering because the mode excitation for this technique is the temperature, i.e., all profiles are excited. Nevertheless, the more complex profile show lower intensity.

For physical insight, we compare the obtained numerical results to the model of Gareeva et al. [27]. Although this approach involves rough approximations on the static configuration, it provides a classification of the modes according to their profile. Gareeva et al. start by estimating the core radius $R_{0}$ corresponding to in-plane magnetization by minimizing the energy
$$E=2\pi t\int_{0}^{R}\left(A\left({\left(\frac{d\theta }{dr}\right)}^{2}+\frac{\sin^{2}\theta }{r^{2}}\right)-D\left(\frac{d\theta }{dr}+\frac{\cos\theta \sin\theta }{r}\right)+K\sin^{2}\theta \right)rdr,$$ where *R* is the dot radius, *t* is its thickness, and $\tan\frac{\theta }{2}=r/R_{0}$. The energy variation with core radius $R_{0}$ for different dot sizes is shown in Figure 10. The calculated core size is about two thirds of the dot size. After calculating $R_{0}$, the static configuration is simplified as θ=0 for $r<R_{0}$ and θ=π for $r>R_{0}$. Thanks to this simplified configuration, Gareeva et al. are able to analytically derive eigenmodes. The radial profile is then a combination of Bessel functions. The azimuthal dependency is exp(imφ). The modes are labelled according to the number of marked maxima of the radial profile *n* and to the azimuthal number *m*. The profiles calculated by MuMax are compared to those deduced from the approach of Gareeva et al. in Figure 9. The main difference lies in the oscillation amplitude in the vicinity of the edge. This discrepancy is due to different pinning conditions at the edge, with no pinning for MuMax calculations and full pinning for the model of Gareeva et al. Except for this difference the profiles are very similar, and the analysis of Gareeva et al. provides a clear method for identifying the eigenmodes.

**Figure 8 materials-15-07474-f008:**
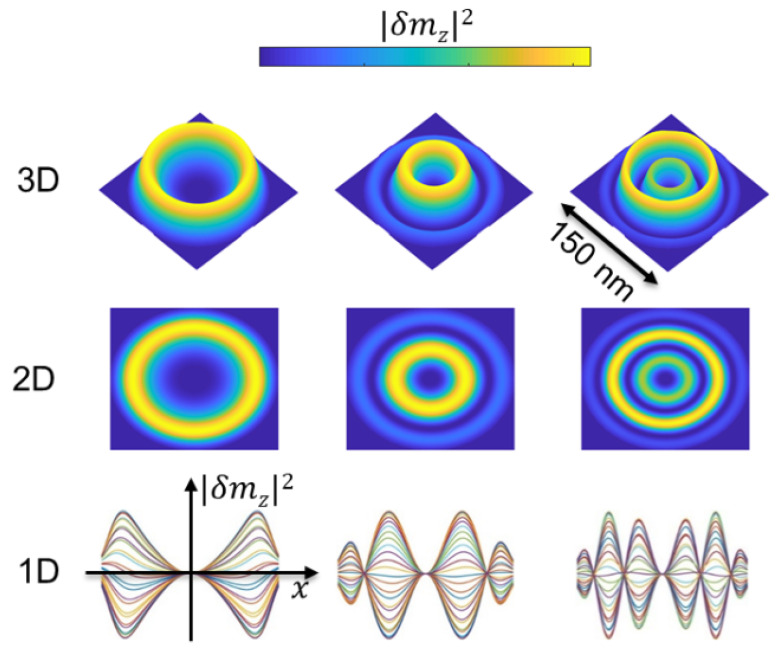
Mode profiles corresponding to eigenfrequencies identified in Figure 6.

**Figure 9 materials-15-07474-f009:**
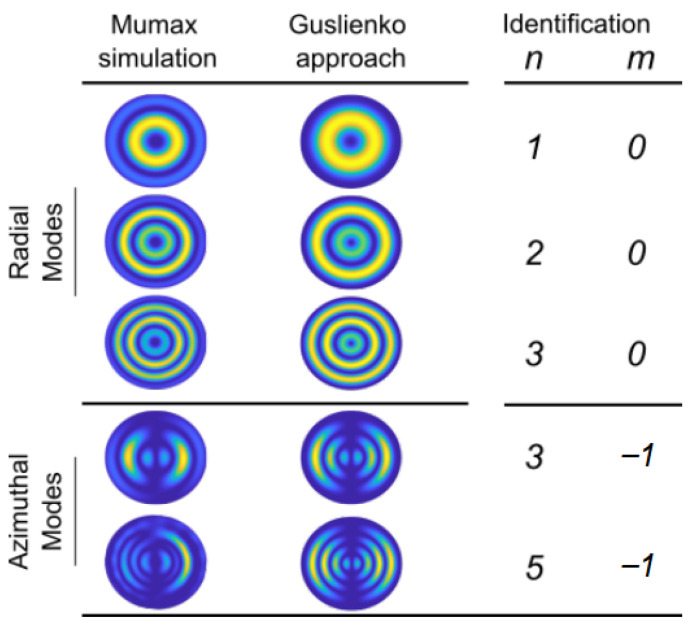
Identification of profile radial and azimuthal modes using the approach of Gareeva et al.

**Figure 10 materials-15-07474-f010:**
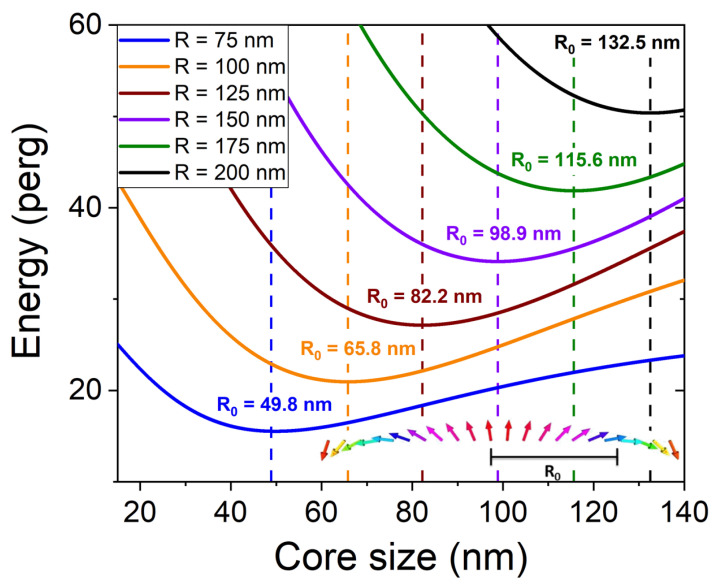
Energy vs. core radius for different dot radii. The calculation parameters are: dot thickness = 1.4 nm, M=1220 emu/cm${}^{3}$, K=−1.04 erg/cm${}^{3}$, D=−1.34 erg/cm${}^{2}$, A=1.6
μerg/cm.

## 4. Conclusions

Stabilization of skyrmions requires a specific set of parameters. The effective anisotropy should be low to ensure that the DMI energy overcomes the exchange energy involved in a domain wall. Moreover, our calculations with our measured DMI constant show that the effective anisotropy should be slightly negative in nanometric dots. Thanks to experimental studies, we identified a real structure that is patterned to host skyrmions. Numerical calculations allowed us to define dot sizes in such a way that eigenmodes could be observed by means of Brillouin light scattering. Moreover, we were able to estimate the duration of the Brillouin light scattering experiment by calculating the power spectral density.

## Figures and Tables

**Figure 1 materials-15-07474-f001:**
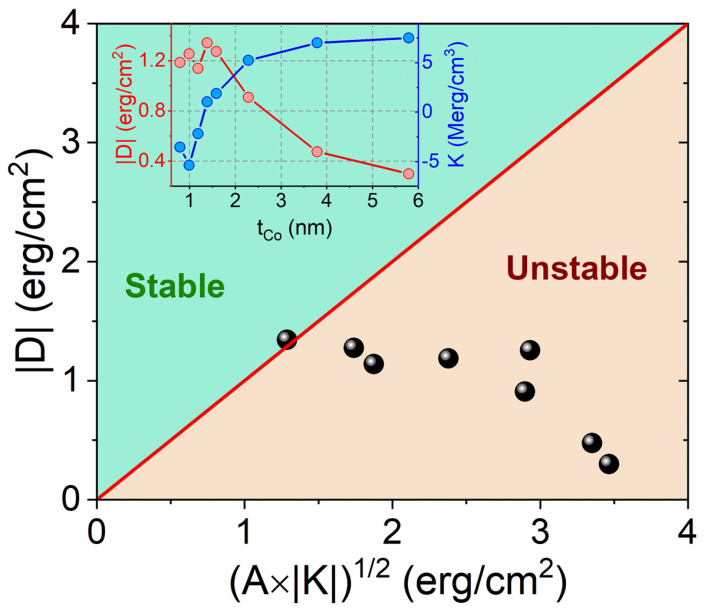
Variation of |D|vs$\sqrt{A|K|}$ for Pt/Co/MgO samples with varying Co thicknesses. The inset shows the Co thickness dependence of |D| and *K* as measured by the Brillouin light scattering technique [28].

**Figure 2 materials-15-07474-f002:**
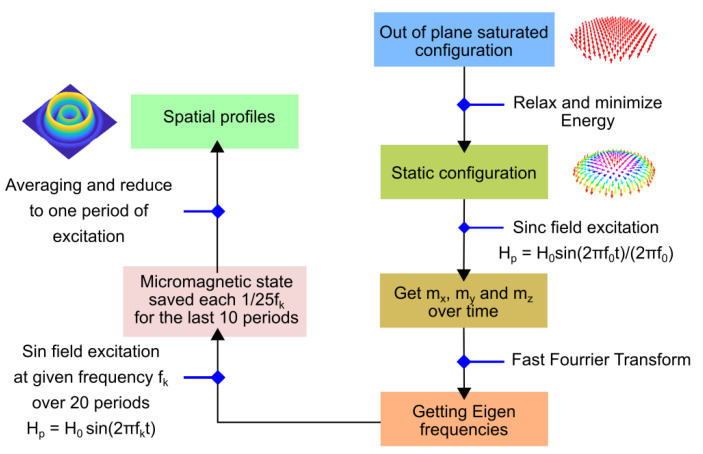
Scheme showing the different steps (1–11) used to obtain the spatial mode profiles.

## Data Availability

Not applicable.

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
