# Peer review of "Theoretical Investigation of Skyrmion Dynamics in Pt/Co/MgO Nanodots"

_materials, 2022, doi:10.3390/ma15217474_

Round 1

Reviewer 1 Report

The authors of the manuscript decided to raise interesting issues related to the theoretical investigation of skyrmion dynamics in Pt/Co/MgO nanodots. The authors prepared a manuscript that may be of use to scientists dealing with identifying the appropriate multilayer in a set of Pt/Co/MgO structures with different Co thicknesses.

It seems to me that there are very few scientific papers of this type that have been published. I am curious, did you find similar research recently carried out by scientists?

The presented data are reliable and useful.

There is no shadow of a doubt that the scientific quality of the manuscript is good. I consider that the figures and tables included in the manuscript are of good quality and require no corrections. The authors correctly included references to the cited works. The English used in the manuscript is enough to be correct and readable.

Summing up, it can be stated that this article is within the scope of Materials. The title is satisfying. The abstract covers appropriate points. The scientific quality of the paper is good. Language doesn't need improvement. Conclusions are adequately supported by the data.

I am convinced that the paper can be published.

Author Response

We thank very much Reviewer 1 for appreciation of our paper.

Reviewer 2 Report

Reports of materials-1974166

Magnetic skyrmions have attracted much research due to their high density, low power consumption, and topologically-protected structures. It is found that many magnetic materials (ferromagnets, anti-ferromagnets, and ferrimagnets) can support skyrmions. One is the Pt/Co/MgO thin film, in which the Dzyaloshinskii Moriya (DM) interaction can be substantial. Hence, the study of the stabilization and the eigenmodes of magnetic skyrmions, which are the central results of this manuscript, is exciting and helpful for people working in this field. Hence, I will recommend the publication of this manuscript if the authors can adequately solve my comments and questions below.

1. The authors should clarify the Hamiltonian (energy) they used to model the Pt/Co/MgO thin film rather than only giving the interaction parameters. Specifically, I would like the authors clearly show the form of anisotropic interaction, DM interaction (bulk or surface), and dipole-dipole interaction.

2. I do not know whether the authors have considered the dipole-dipole interaction (demagnetic field) in their simulations. They used a formula given by Gareeva et al. to explain their numerical results, where the demagnetic field is ignored. This approximation is not guaranteed to be correct since, unlike anti-ferromagnets, the dipole-dipole interaction may be in the same order as the other interactions (e.g., anisotropic and DM interactions) and be non-ignorable. I ask the authors to estimate the magnitude of dipole-dipole interaction and clarify how it affects their results.

3. There is one highly-cited paper [X. S. Wang et al. A theory on skyrmion size, Commun. Phys 1, 31 (2018)] on the size and the stablization of ferromagnetic skyrmion, where a double domain wall profile of skyrmions is suggested. Hence, in principle, one should use two parameters, the radius of skyrmions and the weight of the domain wall, to correctly describe the skyrmion profile. I ask the authors to check whether the simulated skyrmions can also be described by the double domain wall profile.

4. Indeed, the condition of a stable magnetic skyrmion estimated by the authors has been predicted by the above paper [X. S. Wang et al. A theory on skyrmion size, Commun. Phys 1, 31 (2018)], see Eq (14). I ask the authors compare their numerical results with the theoretic prediction.

Author Response

We thank very much Reviewer 2 for giving advices to improve our paper.

1. The authors should clarify the Hamiltonian (energy) they used to model the Pt/Co/MgO thin film rather than only giving the interaction parameters. Specifically, I would like the authors clearly show the form of anisotropic interaction, DM interaction (bulk or surface), and dipole-dipole interaction.

We use the following conventions: z axis is perpendicular to the magnetic layer, x and y axes are in-plane, M is the magnetization, KMC is the anisotropy energy (regardless shape anistropy), Hdem is the demagnetizing field, A is the exchange constant, D is the effective DMI constant and H is the external field. The energy per volume unit of magnetic material reads as

E = – H.M – KMC (Mz/M)2 + (A/M²) ( (∇Mx)² + (∇My)² + (∇Mz)² ) – (1/2) Hdem.M + (D/M²) (M.∇Mz - (∇.M) Mz)

Hdem is determined from Maxwell equations ∇. (Hdem+4πM) = 0, ∇ × Hdem = 0 and the boundary conditions.

This details have been added in the section concerning numerical method.

2. I do not know whether the authors have considered the dipole-dipole interaction (demagnetic field) in their simulations. They used a formula given by Gareeva et al. to explain their numerical results, where the demagnetic field is ignored. This approximation is not guaranteed to be correct since, unlike anti-ferromagnets, the dipole-dipole interaction may be in the same order as the other interactions (e.g., anisotropic and DM interactions) and be non-ignorable. I ask the authors to estimate the magnitude of dipole-dipole interaction and clarify how it affects their results.

We are thankful to Reviewer 2 for raising this important point. MuMax calculations use the energy with no approximation. That is the reason why the discussion is based on MuMax simulations. Nevertheless, as a first approximation and to have physical insight, at the first stage, we have used some approximations. The Rohart et al. criterion involves an approximation for the demagnetizing contribution. This contribution is taken into account as a shape anisotropy i.e. – (1/2) Hdem.M is replaced by 2π(Mz)² . This approximation is also used in the Gareeva calculations. This approximation is rough but it allows to derive a simple criterion. We use this simple criterion as a first guess to select one sample. Then the real magnetization configuration is calculated with the real demagnetizing field derived from Maxwell equations. The Gareeva calulations of eigen modes are compared to the MuMax simulations. They allow for classifying the observed modes according to their profiles. But the real eigen modes are calculated with the real demagnetizing field derived from Maxwell equations.

A sentence clarifying this has been added in the section concerning the numerical results.

3. There is one highly-cited paper [X. S. Wang et al. A theory on skyrmion size, Commun. Phys 1, 31 (2018)] on the size and the stablization of ferromagnetic skyrmion, where a double domain wall profile of skyrmions is suggested. Hence, in principle, one should use two parameters, the radius of skyrmions and the weight of the domain wall, to correctly critical DMI energy constantdescribe the skyrmion profile. I ask the authors to check whether the simulated skyrmions can also be described by the double domain wall profile.

With the parameters (measured on real samples) and considering the experimental protocol where the sample is initially saturated out-of-plane, the only configuration is the one presented in the paper. Other configurations might exist and it is an avenue of investigation in future research but the aim of the paper is to prepare an experimental study. In this study, the sample is initially saturated out-of-plane. This ensures the magnetization configuration is known. If we were sure to stabilize another configuration with another experimental protocol, we would consider it. Here we are sure to stabilize skyrmion with one wall by initially saturating the sample.

Sentences explaining the protocol have been added in sections concerning the numerical method.

4. Indeed, the condition of a stable magnetic skyrmion estimated by the authors has been predicted by the above paper [X. S. Wang et al. A theory on skyrmion size, Commun. Phys 1, 31 (2018)], see Eq (14). I ask the authors compare their numerical results with the theoretic prediction.

In the paper [X. S. Wang et al. A theory on skyrmion size, Commun. Phys 1, 31 (2018)], the demagnetizing contribution is approximated by a shape anisotropy. It is the same approximation as the one considered by Rohart et al. . That is why equation 14 in [X. S. Wang et al. A theory on skyrmion size, Commun. Phys 1, 31 (2018)] yields the same critical DMI energy constant as in Rohart et al. article. As discussed, in the paper this approximation is rough and cannot explain why a positive difference between anisotropy contribution and shape anistropy does not allow skyrmion stabilization.

The reference had been added in the discussion on the difference between taking demagnetizing contribution as shape anisotropy and calculating it from Maxwell equations.

Reviewer 3 Report

In the paper entitled by “Theoretical investigation of skyrmion dynamics in Pt/Co/MgO nanodots”, the authors present a numerical study on stabilization and eigen modes of the skyrmion chiral spin texture in nanodots. The manuscript is well-organized. However, some revisions are required before its acceptance.

1. The abstract section needs polish, especially the last two sentences “Numerical calculations allowed to refine the stabilization conditions. They also allowed to estimate the eigen frequencies and the required accumulation duration for the eigen mode observation by means of Brillouin light scattering”. Please state the novelty of your work, not that of “numerical calculations”.

2. The fonts of text are quite different in each figure. Please keep them uniform throughout the whole manuscript. Arial or Times New Roman may be appropriate.

3. There exist some strange border lines, such as Figure 2, 3, 8, possibly due to screenshot operation. These problems should be avoided in an academic paper.

Author Response

We thank very much Reviewer 3 for giving advices to improve our paper.

1. The abstract section needs polish, especially the last two sentences “Numerical calculations allowed to refine the stabilization conditions. They also allowed to estimate the eigen frequencies and the required accumulation duration for the eigen mode observation by means of Brillouin light scattering”. Please state the novelty of your work, not that of “numerical calculations”.

The abstract has been amended.

2. The fonts of text are quite different in each figure. Please keep them uniform throughout the whole manuscript. Arial or Times New Roman may be appropriate.

 Fonts have been changed in the figures to make them uniform.

3. There exist some strange border lines, such as Figure 2, 3, 8, possibly due to screenshot operation. These problems should be avoided in an academic paper.

Figures have been modified according to reviewer suggestion.

Round 2

Reviewer 2 Report

The authors have solved all the comments I raised. Now, I can recommend the publication of the manuscript.

Author Response

Thank you for your suggestions to improve the paper.